# Understanding the Mechanics and Dynamics of Memorisation in Large Language Models: A Case Study with Random Strings

## Abstract

Understanding whether and to what extent large language models (LLMs) have memorised training data has important implications for the privacy of its training data and the reliability of its generated output. In this work, we focus on the more foundational question of *how LLMs memorise training data*. To this end, we systematically train LLMs of different sizes to memorise *random token strings* of different lengths and different entropies (i.e., sampled from different alphabet distributions) and study their ability to recall the strings. We observe many striking memorisation dynamics including (i) *memorisation in phases* with the alphabet distributions in the random strings being learnt before their relative positions in the string are memorised and (ii) *memorisation in parts* at the granularity of individual tokens, but not necessarily in the order in which they appear in the string. Next, we investigate memorisation mechanics by checking to what extent different parts of a token's prefix in the string are necessary and sufficient to recollect the token. We leverage our insights to explain the dynamics of memorising strings and we conclude by discussing the implications of our findings for quantifying memorisation.

## 1 Introduction

The potential for large language models (LLMs) to memorise training data has many important implications. For instance, memorisation poses serious privacy concerns as it may enable an attacker to establish whether an information instance is part of the training dataset (Biderman et al., 2023a) or to use carefully crafted prompts to cause the generative model to reproduce private information in the training data (Carlini et al., 2021). On the other hand, memorisation may help with establishing factual correctness of generated text (Khandelwal et al., 2019) or assigning authorship / copyright credentials for generated text (Petroni et al., 2019; Reisner, 2023). A flurry of recent research has studied the problem of determining whether or to what extent LLMs have memorised training data (see related work below), but LLM memorisation *mechanisms* still remain poorly understood.

Our focus in this work is to develop *a more foundational understanding of how LLMs memorise training data*. We take inspiration from human learning, where memorisation is viewed as the process undertaken in order to store some information in memory for later recall (Baddeley et al., 2015). The process often involves repeated exposure to the same information to ensure that it can be recollected exactly. In the context of LLMs, the information that can be memorised and recollected are strings of tokens. So our basic approach is to repeatedly train LLMs over token strings and examine how well the LLMs can recollect the token strings.

We study both the *dynamics and mechanics* of LLM memorisation. Under dynamics, we investigate how the ability of LLMs to recollect a string evolves over repeated training iterations over the string. We are particularly interested in understanding whether the ability to recollect the string improves steadily or in phases and whether some parts of the string are memorised before others. Under mechanics, we probe the extent to which different parts of a token's prefix are necessary and sufficient to recollect the token. If an LLM does not need a token's full prefix and can use a partial local prefix to recollect it, it suggests that the LLM has latently stored a mapping of the partial local prefix to token. We use this understanding of the mechanics to explain their dynamics.

To conduct our study, we systematically constructed *random strings* of different lengths (from 16 to 1024 tokens) and of different entropies (by sampling tokens at each position uniformly from different alphabet sets ranging from 2 to 26 characters). We study the ability of LLMs to memorise these random strings via repeated training iterations; for which we use LLMs of different sizes (from 140 million to 12 billion parameters). Our choice of random strings to study memorisation is deliberate. Recollecting tokens that appear at different positions in a random string can only be explained as a result of rote learning and cannot be explained by other factors such as learning rules of grammar or reasoning. Additionally, it is highly unlikely that our pre-trained models would have encountered exact same random strings in their training data.

While restricted, our experimental setup yields several striking insights about memorisation by LLMs. We summarize the major takeaways as follows:

**1. Memorisation happens in phases:** We find two clearly distinguishable phases in the memorisation process: in the first phase (which we call *guessing phase*), LLMs *learn the global context* for the random string, i.e., the alphabet distribution sampled to construct the random string. In the second phase (*memorisation phase*), LLMs *memorise the local contexts* for the random string, i.e., the exact tokens that appear in different positions of the string. The first phase is typically short, while the second phase is longer and varies considerably with string length and alphabet distribution.

**2. Memorisation happens in parts:** We find that memorisation happens *at the level of individual tokens and in no particular order*. That is, during repeated iterations, tokens at a particular position may be memorised before the tokens that appear in positions before or after it.

**3. The role of local prefix in recollection:** When recollecting a token at a given position, we find that a small number of tokens immediately preceding the token, i.e., *the local context*, is sufficient. With repeated iterations, the length of the local prefix needed for recollection decreases. Furthermore, the varying difficulty with memorising different random strings by different models can be explained by the lengths of local prefixes needed to recollect tokens in the string.

**4. The role of full prefix in recollection:** When recollecting token at a position, we find that the exact full prefix is not needed, however, the full prefix needs to *preserve the global context*. That is, recollection would not be affected by a different sampling of tokens from the alphabet distribution in the full prefix, as long as a small length of the local prefix is maintained exactly.

We leverage our insights about LLM memorisation mechanisms to explain the differences in the effort (i.e., the number of iterations) needed to memorise random strings in different settings. We argue that our findings call into question an implicit assumption made by existing studies: that memorisation can be reliably detected by checking if the full prefix up to some position in the string can correctly recollect the full suffix following the position (Biderman et al., 2023a).

## 1.1 RELATED WORK

The topic of memorisation has received great attention in the context of LLMs that are trained on large "internet-scale" data (Song & Shmatikov, 2019; Carlini et al., 2019; Zhang et al., 2021; Biderman et al., 2023a; Mattern et al., 2023; Lukas et al., 2023; McCoy et al., 2023). Most of these works propose a definition of memorisation to test whether the model can generate a given string (present in the training data) using particular prompts or prefixes. While they subtly differ in how exactly they operationalize a measure of memorisation, at a higher level, all these works are concerned with answering the "why" question around memorisation, *e.g.* why should memorisation be a practical concern? To this end, these works show compelling examples of cases where memorisation can hurt (*e.g.* privacy leaks via reconstruction (Carlini et al., 2021) or membership inference (Mattern et al., 2023)). Similarly, there is also a case to be made for memorisation being desirable in cases where the goal is to generate facts and reduce LLM hallucinations. Grounding the generation by LLMs in some verified training data sources can be an effective way to generate trustworthy information (Li et al., 2023; Borgeaud et al., 2022; Khandelwal et al., 2019; Tay et al., 2022; AlKhamissi et al., 2022; Petroni et al., 2019; Guu et al., 2020; Haviv et al., 2022).

We differ from existing works in two key aspects. Firstly, our key goal is to build a foundational understanding of *how* these models memorise. Thus, we do not engage with the question of memorisation being desirable or undesirable and rather provide a mechanistic lens on *how* memorisation happens at an input-output level. Secondly, prior works are motivated by applications and thus simu-

late scenarios where memorisation happens *unintentionally*, *i.e.*, these works typically do not repeat token sequences during training or finetuning (Tirumala et al., 2022; Carlini et al., 2022). We instead force the model to memorise random strings by training on the same tokens multiple times, until the model can generate these random strings verbatim. Our work adds to the nascent literature focused on building a better scientific understanding of memorisation in LLMs (*e.g.* (Tirumala et al., 2022; Jagielski et al., 2022; Carlini et al., 2022; Kharitonov et al., 2021)).

## 2 PRELIMINARIES AND EXPERIMENTAL SETUP

In this section, we provide the necessary notation and concepts that we use throughout the paper. We also describe our experimental setup.

### 2.1 PRELIMINARIES

Throughout, we use random strings in order to train and test LLMs. To create a random string, we first choose an *alphabet* $A$ that consists of $|A| = \ell$ unique tokens; we call $\ell$ the *length of the alphabet*. The alphabet we use for string generation is a subset of a much larger *vocabulary of all tokens* $V$, $A \subset V$. Tokens in a LLM's vocabulary can range from single characters to entire words and its size spans from tens of thousands to a few hundred thousand tokens. In this work, we use tokens corresponding to lowercase characters in the Latin alphabet (see Section 2.2).

We use $P_A$ to denote a probability distribution over the unique tokens of the alphabet $A$. We can compute the *entropy* of $P_A$ using the standard definition of entropy : $H(P_A) = -\sum_{a \in A} P_A(a) \log P_A(a)$. For any given alphabet of length $\ell$ we use $H_\ell$ to denote the entropy of the uniform probability distribution over the alphabet's tokens. We generate a random string $s = (s_1, \ldots, s_n)$ of length $n$ by sampling every token $s_i$ independently from $P_A$. Unless otherwise noted, we assume that $P_A$ is the uniform probability distribution over the tokens in $A$. Given a string $s$ of length $n$ we use $s_{[i,j]}$, with $i \leq j$, to denote the substring of $s$ that consists of positions $(s_i, s_{i+1}, \ldots, s_{j-1}, s_j)$.

Given a string $s$, we train a causal, *i.e. autoregressive*, language model $\mathcal{M}$ to memorise $s$. We do so, by minimizing the *cross-entropy loss* over all positions of $s$. We define the cross-entropy loss of a model $\mathcal{M}$ on string $s$ as follows:

$$\text{Loss}(\mathcal{M}, s) = -\frac{1}{n} \sum_{i=1}^{n} \sum_{t \in V} \delta(s_i = t) \log P_{\mathcal{M}}(s_i = t \mid s_{[1,i-1]}), \tag{1}$$

where, $P_{\mathcal{M}}(s_i = t \mid s_{[1,i-1]})$ is the probability that model $\mathcal{M}$ assigns to token $t$ at the $i$-th position of string $s$; $\delta(\text{condition})$ is an indicator variable that takes value 1 when the condition is true and value 0 otherwise.

Given a string $s$ and a model $\mathcal{M}$, we assume that $\mathcal{M}$ predicts for position $i$ the token with the largest $P_{\mathcal{M}}(s_i = t \mid s_{[1,i-1]})$). Then, we define the *recollection accuracy* of $\mathcal{M}$ with respect to $s$ based on greedy decoding as:

$$\text{Accuracy}(\mathcal{M}, s) = \frac{1}{n} \sum_{i=1}^{n} \delta(s_i = \arg\max_{t \in V} P_{\mathcal{M}}(s_i = t \mid s_{[1,i-1]})). \tag{2}$$

### 2.2 EXPERIMENTAL SETUP

**Data generation process:** In our experiments, we focus on alphabets $A$ with $\ell \in \{2, 4, 7, 13, 26\}$. We use tokens corresponding to the first $\ell$ lowercase characters from the Latin alphabet, *i.e.* $A \subseteq \{a, \ldots, z\}$. We therefore often refer to the elements of the alphabet as characters, even though they are technically tokens. We generate random strings of lengths $n \in \{16, 32, 64, 128, \ldots, 1024\}$ by sampling tokens uniformly at random from $A$. All our results are aggregates over 10 runs with different random string samples; we highlight one standard deviation in the plots. We show examples of random strings in Appendix A.4.

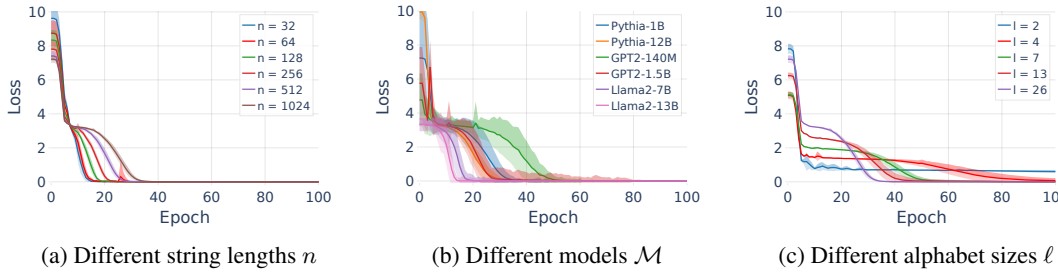

| (a) Different string lengths $n$ | (b) Different models $\mathcal{M}$ | (c) Different alphabet sizes $\ell$ |

Figure 1: **[Training loss.]** Cross-entropy loss curves for memorising random strings while varying the string length $n$, model type and size $\mathcal{M}$, and the alphabet size $\ell$. For all models, the loss initially decreases quickly, before stagnating and then decreasing again until convergence. Models need more epochs to memorize longer strings than shorter ones, *i.e.* it takes longer for the loss to converge to 0 for longer string lengths. Models with a higher parameter count, memorise more quickly, although there seem to be diminishing returns beyond around 1 billion parameters. Memorisation of strings with smaller alphabet sizes initially progresses more quickly, but subsequently stagnates and is slower than for larger alphabets.

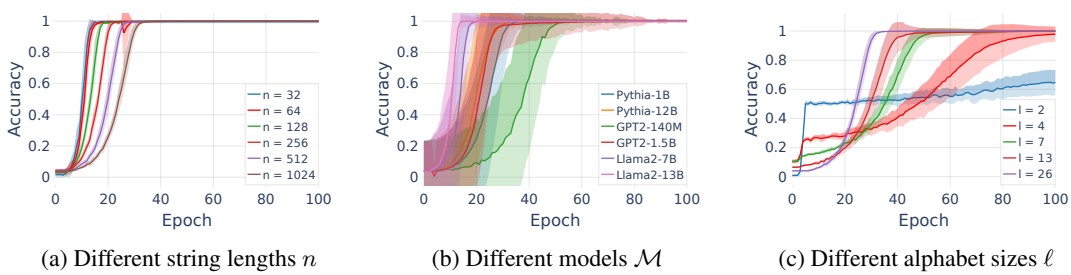

| (a) Different string lengths $n$ | (b) Different models $\mathcal{M}$ | (c) Different alphabet sizes $\ell$ |

Figure 2: **[Recollection accuracy.]** Token recollection accuracy curves for memorising random strings while varying the string length $n$, model type and size $\mathcal{M}$, and the alphabet size $\ell$. Accuracy increases only slowly initially, before accelerating and converging to 1.

**LLM models and training:** We use pretrained models of the Pythia (Biderman et al., 2023b), GPT-2 (Radford et al., 2019) and Llama-2 model (Touvron et al., 2023) families. For the Pythia model family, we use variants with 1B and 12B parameters, for the GPT-2 family we use 140M (GPT-2) and 1.5B (GPT-2-XL) parameter models and for Llama-2 we use 7B and 13B parameter variants. We choose these models to span a wide spectrum of parameters counts, and refer to the respective model by its parameter count, *e.g.* Pythia-12B or GPT-2-1.5B. We train models for 100 epochs to memorise strings. Additional details about the training setup are in Appendix A.1.

## 3 DYNAMICS OF MEMORIZATION

Our investigations in this part is driven by the question: how does an LLM learn to memorise a random string over time? We explore this question along several relevant dimensions.

Throughout this Section, unless stated otherwise, we use $n = 1024$ token strings, alphabet size $\ell = 26$, $\mathcal{M} = $ Pythia-1B, and train models for 100 epochs.

### 3.1 ON THE MEMORABILITY OF DIFFERENT TOKEN STRINGS

**Two phases of memorisation:** Our first striking observation is that, across all string lengths, alphabet sizes and model types — except Llama-2, which we discuss next — in Figure 1, we observe the same convergence pattern of the training loss. Initially the loss decreases quickly, up until around epoch 10, where it plateaus for 10 or more epochs, before undergoing another decline, culminating in convergence. At the same time, the recollection accuracy curves in Figure 2 show that the initial drop in the loss value does not coincide with a rise in accuracy. Rather, accuracy remains low during the initial phase of loss reduction and only increases substantially during the second loss de-

scend phase. The seeming exception to this pattern are the accuracy curves for smaller alphabets in Figure 2c, which initially rise quickly, however, only to the random chance level for the respective alphabet, after which they also plateau and rise slowly, in sync with the second phase of the loss decline.

Based on this consistent pattern, we postulate that there are two distinct phases of memorisation: an initial drop in the model's loss before it enters the plateau region, which we call the *Guessing-Phase*, followed by a transition out of the plateau and a subsequent convergence, matched by an increase in accuracy, which we call the *Memorisation-Phase*. We investigate these two phases more closely in Section 3.2, and discuss our naming choice.

**Llama-2 models skip the *Guessing-Phase*:** The striking exception to the two phases observation are the Llama-2 models. Figure 1b shows that these models only exhibit a single descend phase, and that their initial loss values before any training match those that the other models reach after completing the *Guessing-Phase*. We investigate this phenomenon further in Appendix A.3 and show that — in contrast to the other models — Llama-2 models exhibit strong in-context learning abilities and are able to infer the alphabet distribution $P_A$ with just a few tokens of context. Therefore, they can skip the *Guessing-Phase* and can start directly with the *Memorisation-Phase*.

**Entropy and memorability:** The third interesting observation that we make is that strings with smaller alphabets are harder to memorise than those with larger alphabets. In Figure 1c we see that for smaller alphabets, the loss drops faster initially during the *Guessing-Phase*, but stagnates and decreases more slowly in the subsequent *Memorisation-Phase* than for larger alphabets. The model memorising a string with $\ell = 2$ for example is not able to converge at all during the 100 epochs of training. The corresponding accuracy curves in Figure 2c show that accuracy initially jumps to the random chance level (which is higher for smaller alphabets), but then increases much more slowly for smaller alphabets than for larger ones. In Appendix A.2 we show that this pattern is not specific to the alphabet size, but to the entropy of $P_A$ used to generate the string.

## 3.2 DYNAMICS I: ON THE PHASES OF MEMORISATION

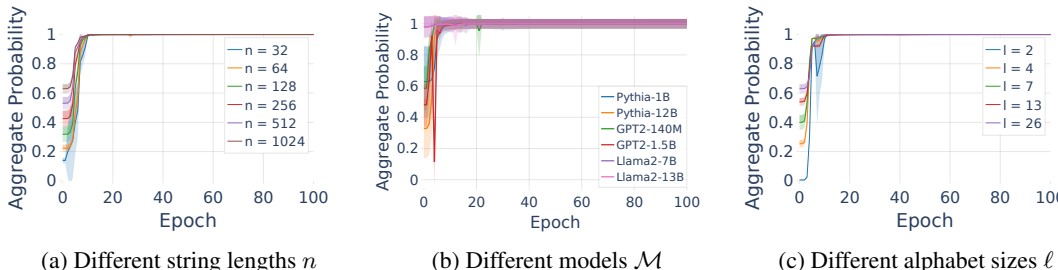

(a) Different string lengths $n$      (b) Different models $\mathcal{M}$      (c) Different alphabet sizes $\ell$

Figure 3: **[Aggregate probability mass over within-alphabet tokens $A$.]** Plots show $\sum_{t \in A} P_{\mathcal{M}}(s_i = t \mid s_{[1,i-1]})$, averaged over all $i$'s. In all cases, models quickly learn to allocate the maximum possible probability mass to the tokens within the alphabet $A$, *i.e.* they only predict tokens from $A$ after a few training epochs.

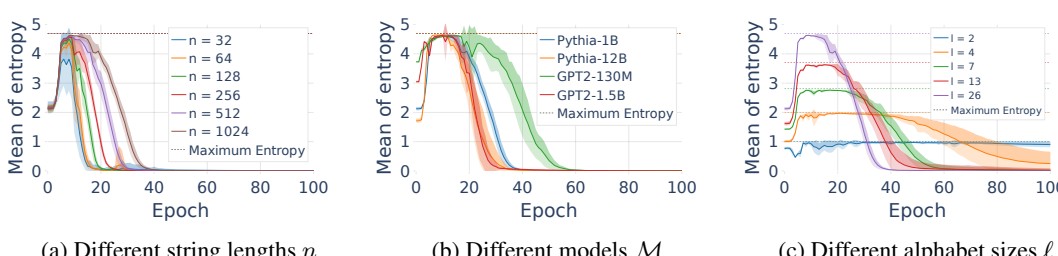

(a) Different string lengths $n$      (b) Different models $\mathcal{M}$      (c) Different alphabet sizes $\ell$

Figure 4: **[Entropy in the models' probability distribution.]** We show the average entropy of the probability distribution of model $\mathcal{M}$, *i.e.*, $P_{\mathcal{M}}(s_i = t \mid s_{[1,i-1]})$ for $t \in V$, averaged over $i$. The entropy initially rises to its maximum value, before decreasing to 0. The maximum entropy that can be attained (for different alphabet lengths) is shown as a reference point.

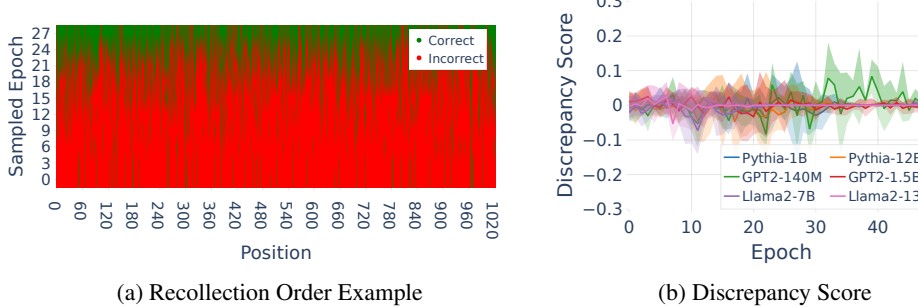

(a) Recollection Order Example          (b) Discrepancy Score

Figure 5: **[Recollection order in the memorisation processes.]** Demonstrating visually (Fig. 5a) and statistically (Fig. 5b) that for strings length $n = 1024$ the positions of recollected tokens are uniformly distributed.

Why are we observing two distinct and consistent phases of memorisation, and what is happening in each of them? To answer this, we take a closer look at the token-level probability distributions produced by the models. Figure 3 shows the aggregate probability mass that models assign to the tokens inside the alphabet $A$, i.e., $\sum_{t \in A} P_{\mathcal{M}}(s_i = t \mid s_{[1,i-1]})$, averaged over all positions $i$. Analogously to the quick initial decrease in loss (Figure 1), we see that models quickly learn to assign all the probability mass to tokens within the alphabet $A$ and separate those from the whole vocabulary of tokens $V$.

We also compute the token-level entropy over the course of training; i.e., the entropy of $P_{\mathcal{M}}(s_i = t \mid s_{[1,i-1]})$ for $t \in V$, averaged over $i$. The results in Figure 4 show a sharp increase in entropy to the maximum possible value in the initial stages of training; that coincides with the rise in aggregate probability mass and the initial drop of the loss curves. After the entropy peaks at around epoch 15, it drops to 0, matching the second decrease of the loss function and the rise in accuracy (Figure 2).

Thus, in the first phase (*Guessing-Phase*), the models are learning which tokens are in $A$ and separate those from $V$. In that phase, they do not know *which specific tokens* to predict at each position in the string and thus *guess* tokens from $A$ randomly. In the second phase (*Memorisation-Phase*), the models become more accurate and they actually memorise the correct tokens at each position.

## 3.3 DYNAMICS II: ON THE MEMORISATION ORDER OF INDIVIDUAL TOKENS

We aim to characterize the *Memorisation-Phase*, more closely and ask: "is there a specific ordering in which tokens are memorised or are the positions of the correctly recollected tokens random?"

Figure 5a shows which tokens in a string have been memorised correctly during different – initial – epochs of training. There is no discernible order to the memorisation. Tokens in the middle or at the end of the string might be memorised before earlier tokens. Additionally, memorisation is not stable in the initial parts of training, until the *Memorisation-Phase* has made some progress; previously memorised tokens are often forgotten (*i.e.* predicted incorrectly) again at later training epochs, until memorisation starts to converge at around epoch 20.

In order to quantify whether the memorised positions are uniformly distributed we compute the *discrepancy score* – an operationalization of the notion of discrepancy in statistics (Niederreiter, 1992) – as follows: for any epoch, we count the number of correct recollections of our model on the *input string* and then we pick uniformly at random the same number of positions on *random string* of the same length. Utilizing a fixed window of 20 tokens, we randomly sample 50 substrings from each of the two strings. For each of these sampled substrings, we calculate difference in the number of correct recollections between the tested and target substrings. The average of these differences provides the *discrepancy score*. The low discrepancy score observed in Figure 5b suggests that for all the models evaluated, the memorised positions are random.

Thus, we conclude that memorisation happens at the granularity of individual tokens and not entire strings. Also, memorisation of individual tokens does not necessarily happen in the order in which they appear in the string, i.e., tokens at a particular position of a string can be recollected even when tokens preceding or succeeding them cannot.

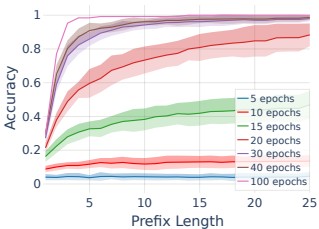 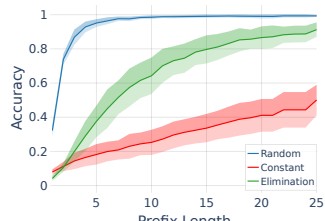 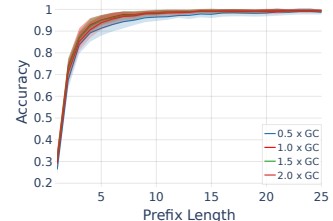

(a) Different training stages

(b) Changing the global context distribution

(c) Different amounts of global context

Figure 6: **[Recollection accuracy for different prefix lengths.]** We test models with inputs where we keep short local prefixes immediately preceding the target token fixed, while replacing all other tokens according to different replacement strategies and global context lengths that are multiples of the original length (GC).

# 4 MECHANICS OF MEMORISATION

Our previous analysis does not answer *how* LLMs regurgitate tokens shown during training. Here, we look at what parts of the string, local (*i.e.* just a few preceding tokens) or global (*i.e.* all preceding tokens) are necessary to recollect a particular token. We call this part *mechanics*, because — similar to physics — our findings can explain and predict some of the dynamics we observe.

## 4.1 EXPERIMENTAL SETUP

Given a token at position $i$ in random string $s$ sampled from $P_A$ as described in Section 2.2, we can split $i$'s full prefix $s_{[1,i-1]}$ into two parts: a *local prefix* of size $k$, *i.e.* $s_{[i-k,i-1]}$, and the *global context*, given by $s_{[1,i-k-1]}$. In our analysis, we keep the local prefix fixed while modifying the global context. In doing so, we can determine how much information we need to retain, and how much we can change, in order to still correctly recollect token $i$. We change the global context using one of the following *replacement policies*: (a) Random, (b) Constant and (c) Elimination. Random replaces every token of the global context with a different one sampled randomly from $P_A$. Constant replaces every token in the global context with a fixed token from $A$. Finally, Elimination removes the global context completely by masking it out.

For the experiments in this section we use a Pythia-1B model that is memorising an $n = 128$ string with $\ell = 26$. Unless, otherwise noted, we use the Random replacement policy for generating the global context. We also report the recollection accuracy at epoch 100 – *i.e.* after convergence.

## 4.2 MECHANICS I: ON THE ROLE OF LOCAL PREFIXES

To determine how much information we need to retain in order to recollect memorised tokens, we test models with different local prefix lengths, while applying Random to the global context. Figure 6a shows the recollection accuracy for different prefix lengths at different training stages.

From the plot we observe that in the initial *Guessing-Phase* (at epoch 5 and 10), all prefix lengths achieve very low accuracy. Starting from epoch 15 (when the training transitions into the *Memorisation-Phase*), the accuracy starts to increase substantially, even for short prefixes. Notice that especially at epoch 30 and later, the accuracy of short prefixes, which correspond to less than 5% and 10% of the total string length, is close 100%, which is also the performance of the full prefix when the model has converged. Thus, small local prefixes – much shorter than the entire string – are very effective at correctly recalling tokens.

## 4.3 MECHANICS II: ON THE ROLE OF GLOBAL CONTEXT

In order to further investigate the role of global context in token recollection, we consider the rest of the replacement policies we described in Section 4.1.

**Importance of the replacement policy:** Figure 6b shows the accuracy of immediate prefixes with global contexts being generated using all replacement policies. Our results show that when trying

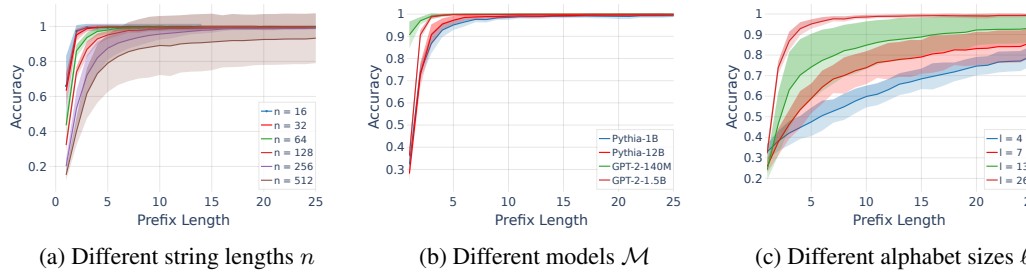

(a) Different string lengths $n$     (b) Different models $\mathcal{M}$     (c) Different alphabet sizes $\ell$

Figure 7: **[Recollection accuracy for local prefixes of different lengths.]**

to recollect tokens from memorised strings, the global context is important. Maintaining the original distribution $P_A$ of the tokens in the global context (even though the substring itself changes), allows for high recollection accuracy. Removing `Elimination` or biasing `Constant` the global context leads the model to assume a wrong distribution, lowering its recollection accuracy. In fact, `Constant` leads to lower accuracy than `Elimination` implying that the wrong distribution in the global context is worse than no distribution at all.

**Importance of the length of global context:** Is it enough to maintain $P_A$ when generating the global context, or does the length of the context also matter? To determine this, we increase (by 100%, 150% or 200%) or decrease (by 50%) the number of tokens in the global context while applying the `Random` replacement strategy. We show in Figure 6c that adding or removing tokens from the global context does not impact the recollection accuracy for fixed local prefix length. Put differently, the amount of global context, and thus the position of $i$ within $s$ is not important, as long as some global context information, and the local prefix, are present.

In summary, while only keeping the tokens in the local prefix constant is mostly sufficient to recollect the target token, this only holds when the token distribution in the global context is preserved.

### 4.4 PUTTING IT ALL TOGETHER: HOW MECHANICS EXPLAIN THE DYNAMICS

The insights about the role of local prefixes and global context, enables us to explain some of dynamics we observed in Section 3. For instance, in Figures 1c and 2c we saw that models are slow to memorise strings with small alphabet sizes. When we compare this trend with the performance of local prefixes on the same data shown in Figure 7c, we can understand why this is happening. Strings with smaller alphabets (*i.e.* with lower entropy) require longer prefixes to achieve the same level of accuracy as for larger alphabets. Therefore, models have to memorise longer and more complex associations in order to correctly predict tokens, which requires more training epochs.

Similarly, Figure 7a shows that longer strings also require longer local prefixes, which is consistent with the slower memorisation speed we observed in Figures 1a and 2a. We do not observe stark differences in the performance of different prefix lengths for different models, shown in Figure 7b, which matches our observations made in Figures 1b and 2b, where we see roughly similar memorisation speeds for different models.

Putting it all together, our model of memorisation in different phases, as well as in parts, in conjunction with the roles of local prefixes and global context provides new prespectives of how LLMs memorise random strings.

## 5 DISCUSSION AND IMPLICATIONS

**Generalising to multiple strings:** To test if some of our findings apply to scenarios with multiple strings, we show in Figure 8 that memorising one long string or multiple shorter partitions of the same string is equally fast. We also show the accuracy of predictions during this process in C.1. While limited, this experimental setting indicates the potential for our findings about LLMs' memorisation mechanisms to generalise to multi-string settings.

**Generalising to non-random strings:** As the notion of alphabet distributions are less well-defined for non-random strings, it is unclear what may be learnt during the first phase of memorisation. On

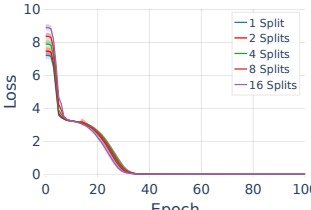 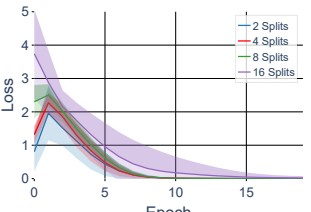 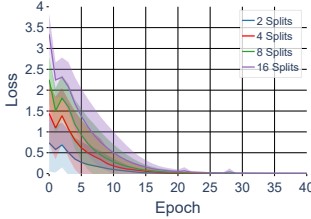

(a) Memorising the same string with different splits

(b) Memorising entire string after memorising splits

(c) Memorising splits after memorising entire string

Figure 8: **[Experiments to support generalization to multiple strings.]** We train Pythia-1B models to memorise a single 1024 length random token string as well as its partition into $p \in \{1, 2, 4, 8, 16, 32, 64\}$ equal-sized substrings. a) The effort needed to memorise the different sets of multiple partitions. The training loss plots show that memorising partitions of a string requires the same effort as memorising the original string. This observation can be explained by the fact that the aggregate number and sizes of local prefixes that need to be memorised would not change with splitting a string. b) (and c) further confirm this observation, by showing that memorising a random full string after memorising one of the splits (and vice-versa) takes just a small number of iterations.

the other hand, one simple way to construct multiple full prefixes that preserve alphabet distribution for non-random strings may be to permute the tokens that are outside the local prefix.

**Generalising to untrained models:** One of our key findings is that pre-trained LLMs learn the alphabet distribution of random token strings in a small number of iterations during the first phase of memorisation. It is worth checking if such efficient learning can happen with untrained models and if memorisation would be equally effective using untrained models. We expect that some of our key findings will generalise to new settings, and even when others don't, our analysis provides a useful framework for future investigations of memorisation mechanisms in LLMs.

**Implications for privacy concerns with memorisation:** Our findings about memorisation in parts and the relative roles of global context and local prefixes in recollecting a token suggest that the potential for membership inference attacks and leakage of private information may be worse than previously anticipated. For instance, to infer whether a token string is a member of the training data, it may be sufficient to infer that some unique part of the string has been memorised. Similarly, if some private information appears repeatedly as part of several token strings, particularly with similar alphabet distributions, its chance of being memorised may be higher.

**Implications for quantifying memorisation:** Our findings suggest that future measures to quantify memorisation of a string may have to start with measuring memorisation at the granularity of individual tokens. Appendix C.2 shows that string-level measures can severely underestimate memorization. However, our findings call for caution against inferring that a token has been memorised because it has been correctly recollected; we showed that tokens in random strings generated using a small alphabet set may be recollected correctly, once the alphabet distribution has been learnt, even though they have not been memorised. In the future, it is worth exploring whether a more reliable test for a token's memorisation might be checking if a token can be recollected correctly using its local prefix of small length.

## 6 CONCLUSIONS

In this paper, we made significant steps towards developing a foundational understanding of how LLMs memorise training data. For this, we studied both the *dynamics* and the *mechanics* of LLM memorisation. The former focuses on how the memorisation evolves over the training time and the latter allows us to identify latent mechanisms by which models recall memorized tokens. In terms of dynamics, our study revealed different phases of memorisation; in terms of mechanics it revealed that both the local prefix and the global context play a role in token recollection and there is an interplay between them. We also showed how the mechanics can explain the dynamics we observed. Although we considered a particular experimental setting, we showed some evidence that it generalizes to other settings. Moreover, the insights this study offers can potentially further refine the quantification of memorisation achieved by LLMs.

## REPRODUCIBILITY STATEMENT

We run all our experiments on publicly available, open-source models and thus all our results can be seamlessly reproduced. We also attach our code to aid reproducibility which includes a README with instructions on how to re-generate all of our results. To ensure correctness, we also report all our results over 10 random seeds. All other details about pre-processing, learning rate, epochs, model architectures, and details on how to generate random strings can be found in the Appendix A& B and are also included in our attached code.

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

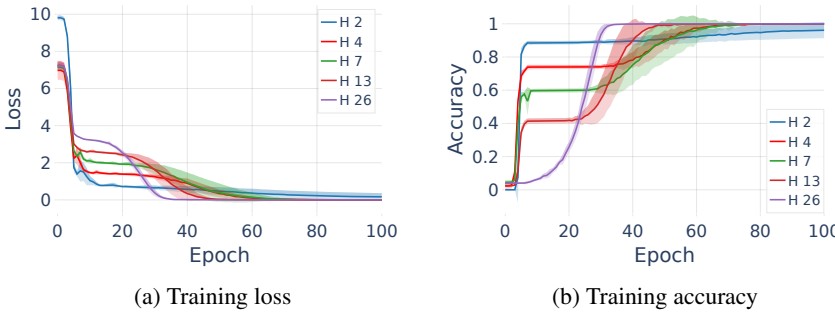

(a) Training loss            (b) Training accuracy

Figure 9: **[Memorisation dynamics when controlling for entropy.]** Loss and accuracy curves for memorising strings with $n = 1024$ tokens by a Pythia-1B model, created from an alphabet with $\ell = 26$. We control the entropy level of the strings by changing the probability of the first character ("a") in the alphabet, such that the entropy becomes the same as those of the smaller alphabets with 2, 4, 7 and 13 characters, *i.e.* $H_2$, $H_4$, $H_7$ and $H_1 3$. We observe dynamics that are very similar to those of different alphabet sizes with the same entropy level in Figure 2c and Figure 1c

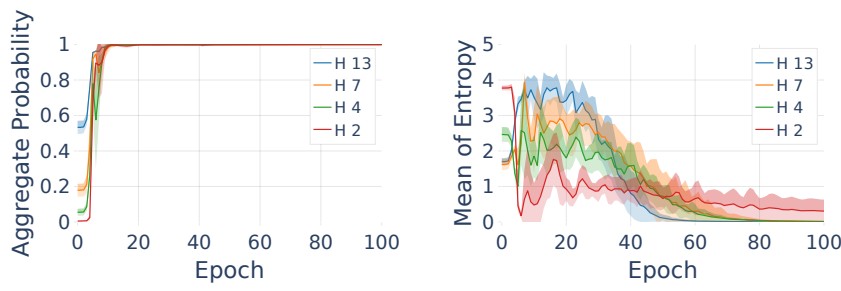

(a) Different entropy levels for the 26 character alphabet     (b) Different entropy levels for the 26 character alphabet

Figure 10: **[Aggregate probability mass and entropy when controlling for entropy.]**

# A    ADDITIONAL DETAILS ON THE DYNAMICS

## A.1    TECHNICAL DETAILS ON THE TRAINING SETUP

**Models:** In this paper, we use pretrained models of the Pythia (Biderman et al., 2023b) and GPT-2 (Radford et al., 2019) family. For the Pythia model family, we use variants with 1B and 12B parameters, and for the GPT-2 family we use 140M (GPT-2) and 1.5B (GPT-2-XL) parameter models. Pretrained versions for all models are publicly available on the Huggingface Model Hub.

**Data:** We fine-tune models on random strings for 100 epochs, with a linearly decaying learning rate schedule, with 5 warmup steps/epochs (for single strings, each step is an epoch). For the Pythia models we use a maximum learning rate of $10^{-5}$, for GPT-2-140M we use $10^{-3}$, and for GPT-2-1.5B we use $10^{-4}$. These learning rate values resulted in the fastest convergence during a grid search over values from $10^{-3}$ to $5^{-7}$.

## A.2    ADDITIONAL RESULTS ON THE EFFECT OF STRING ENTROPY

In Section 3.1 we observe in Figure 1c and Figure 2c that models memorise strings with smaller alphabet sizes more slowly. Our conjecture is that these differences in convergence speed might be linked to the strings' entropy; when training on strings with lower entropy (*i.e.* with smaller alphabets), models initially have a large drop in their loss, but after this initial drop, they subsequently learn a lot slower. If this were indeed the case, and memorisation behaviour is linked to entropy,

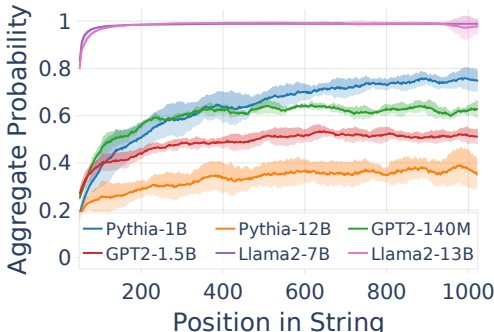

Figure 11: **[Aggregate probability over all alphabet tokens at different string positions before training.]** The aggregate probability over the tokens in the alphabet shows how well models are able to infer $P_A$ from the prefix of the string at a given positions. Models differ in their ability to learn the distribution via in-context learning. Llama-2 models are particularly good, assigning almost all probability mass to tokens in the alphabet with just 100 tokens of context. Other models do not exhibit the same in-context learning abilities. The results show $n = 1024$ token strings over the $\ell = 26$ character alphabet.

then we should expect similar patterns to those in Figures 1c and 2c to hold for other strings with the same levels of entropy.

To test whether this is the case, we lower the entropy of the 26 character alphabet strings by biasing the sampling of tokens towards the first character ("a"), such that the entropy is lowered to the level of the smaller alphabet sizes.

And indeed, Figure 9 shows a very similar trend for lower entropy versions of the $\ell = 26$ alphabet. Initially, the loss decreases quickly, while accuracy jumps to the random chance levels of guessing "a", $0.41, 0.60, 0.75, 0.89$ for the 13, 7, 4 and 2 character alphabets, respectively. Afterwards, learning is much faster on higher entropy distributions. These results show that the memorability of a random string is inherently tied to its entropy, and that our observations about different alphabet sizes are in fact more general.

In Figure 10 we also observe similar trends for the aggregate alphabet probability as well as the entropy that the model assigns to the string as in Figure 3c and Figure 4c.

## A.3 ADDITIONAL RESULTS ON IN-CONTEXT LEARNING

We saw in Section 3.1 that all models exhibit two phases of memorization, except the Llama-2 models, which skip the *Guessing-Phase* and start directly with the *Memorisation-Phase*. To better understand why this is happening, we test how well the different models are able to learn the distribution of the random strings $P_A$ via in-context learning.

In Figure 11 we show the aggregate probability over all tokens in the alphabet that models assign at each position in the string. We use a sliding window of size 50 to smooth the curves. The figure shows models at epoch 0, *i.e.* before they have started to memorize the string. Thus, without any training, models can only infer the string distribution via in-context learning.

Indeed, we see that Llama-2 models exhibit strong in-context learning abilities and quickly assign all probability mass to tokens within the alphabet. The other models are not able to infer the distribution nearly as well. Note that the differences in in-context learning ability observed in Figure 11 correspond to the differences in the initial loss in Figure 1b. Thus, the *Guessing-Phase* appears to be a stage that all models go through, but sufficiently strong in-context learning abilities allow models to effectively shorten it to zero.

| Alphabet and distribution | Tokens |
|---|---|
| 2 characters, uniform | bbabbabbababbabaaabbababaaaababb |
| 4 characters, uniform | cdbccbddbcaddbcabaccbcbcabaacadd |
| 7 characters, uniform | efceecffdeaggdebbbffddbdabaafaff |
| 13 characters, uniform | hleijdjkfibllfhcdcjjghdgbdaajakk |
| 26 characters, uniform | pwjqshtulrcxxlpegessmognchaatauv |
| 26 characters, H2 | aaaaaaaaaagaaaaaaaaaaaaaaaaaaaaa |
| 26 characters, H4 | alaaaabfaaaroaaaaaaaaaaaaaaaaadj |
| 26 characters, H7 | bqadhakmagausabaaaiiaaaaaaaajalp |
| 26 characters, H13 | taknapqbmawvbjaaaoodgafaaaaoaqsa |

Table 1: **[Examples of random strings used in the paper.]** We show the first 32 tokens/characters.

## A.4 Examples of random strings used in the paper

Table 1 shows examples of random token strings used in the paper. Each character is tokenized individually.

## B Additional Details on the Mechanics

### B.1 Additional details on the experimental setup for testing local prefix accuracy

We test whether a token $s_i$ in string $s$ sampled from distribution $P_A$ can be correctly recalled with a local prefix $s_{[i-k,i-1]}$ of length $k$ for the different replacement strategies Random, Constant, and Elimination.

To test recall for the Random strategy, we sample 100 replacements $r_j$ of length $i - k - 1$ for the global context $s_{[1,i-k-1]}$ from $P_A$. Then we compute how many times the model correctly predicts token $s_i$, given the input $r_j \circ s_{[i-k,i-1]}$, *i.e.* when we randomize all tokens in the input according to $P_A$, other than the local prefix. If a *plurality* of predictions among the 100 samples match $s_i$, *i.e.* if $s_i$ is the most frequently predicted token, then we say that the local prefix of size $k$ can correctly recall $s_i$.

For the Constant strategy we follow a similar process, but instead of sampling each token in $r_j$ randomly from $P_A$, we only sample a single token, that we use at all positions in $r_j$, for each of the $r_j$'s separately. For the Elimination strategy, we simply omit the global context, *i.e.* we only use the local prefix $s_{[i-k,i-1]}$ as a single sample.

## C Additional detail on the Implications

### C.1 Additional Results on memorising Splits followed by entire string

Figure 12 shows that memorising the entire 1024 length string is substantially faster after memorising the splits independently. The first split is always perfectly memorised at the start (e.g. the first 512 positions in 12a) however the subsequent partitions are not as they now have additional context behind them during inference and also have different positional encodings. However as can be noted by looking at the initial epochs, strings apart from the first string also have considerable memorised portions which hints towards local prefix based memorisation. At the same time, the tokens at junctions (where splits of strings are connected) are incorrect more often due to new prefix being present which was not present earlier and might be in conflict with some other prefix learnt. Over the course of training the models also forget portions which were earlier perfectly memorised as can be seen in the first epoch but eventually perfectly memorise the string. This forgetting is likely due to the conflicts that arise with similar prefixes and the models being forced to relearn mappings for some positions.

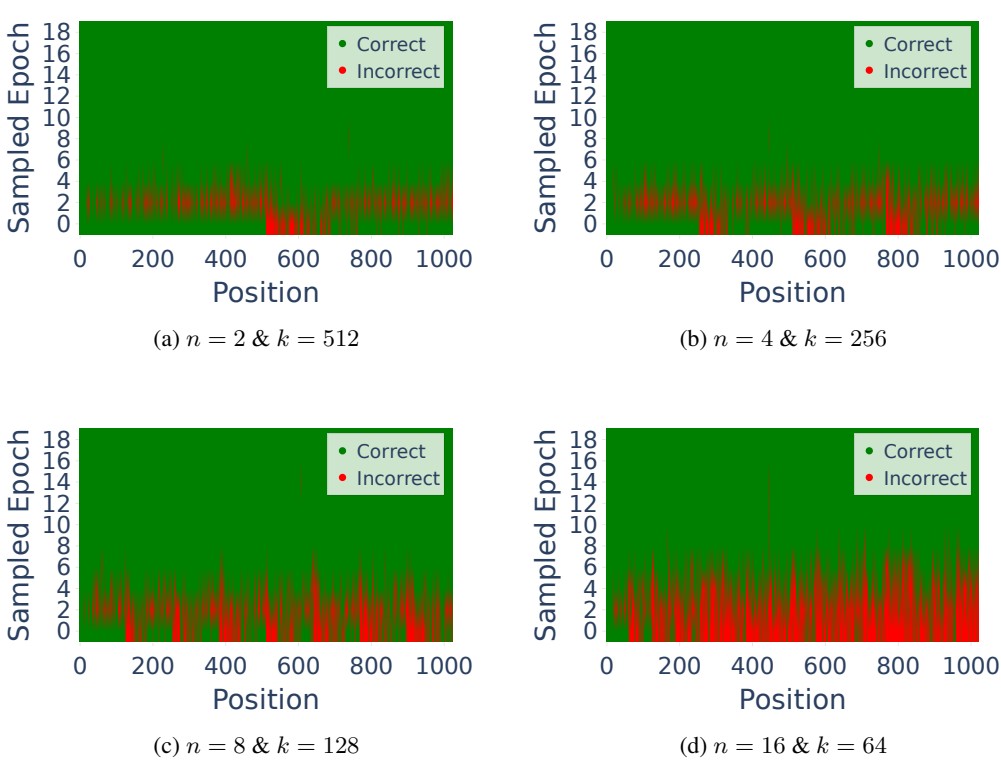

(a) $n = 2$ & $k = 512$

(b) $n = 4$ & $k = 256$

(c) $n = 8$ & $k = 128$

(d) $n = 16$ & $k = 64$

Figure 12: Memorising strings using Pythia 1b model in 2 phases. First split a string of length 1024 into $n$ string of equal length $k$ and memorise those perfectly. Then memorise the original 1024 length string using the trained model.

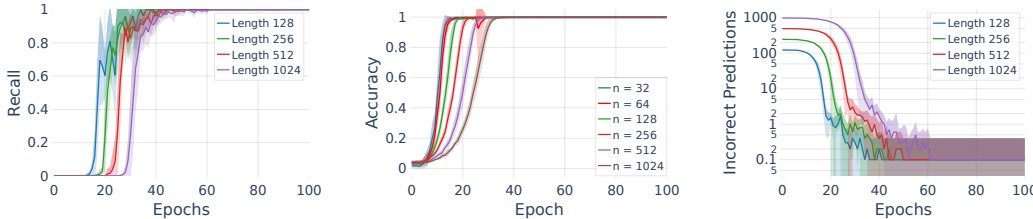

(a) Fraction of string positions re-called as memorised.

(b) Prediction accuracy for different $n$

(c) Number of incorrect predictions at each epoch.

Figure 13: **[Applying string-based memorisation measures to random strings.]** String-based memorisation metrics severely underestimate the degree of memorisation in random strings. The number of substrings detect as memorised with a strict 50 correct adjacent token requirement in (a) is much lower than the prediction accuracy of the model over individual string positions in (b) (same plot as in Figure 2a).

## C.2 EXISTING MEMORIZATION MEASURES CAN SEVERELY UNDERESTIMATE THE DEGREE OF MEMORIZATION

We apply the popular memorization metric from Carlini et al. (2022) in our random token string setting. The measure detects a string $s$ as memorised by model $\mathcal{M}$ if $\mathcal{M}$ produces $s$ (with greedy decoding) when prompted with string prefix $p$, and $[p||s]$ is contained in $\mathcal{M}$'s training data.

We apply this measure to $n = 1024$ random token strings at each training epoch of a Pythia-1B model and measure the fraction of substrings that it detects as memorised. Analogously to Carlini et al. (2022), we set the string length $|s|$ to 50 tokens. Then, we detect for every position $i$ in the 1024 token string whether, when prompted with the entire preceding string, $\mathcal{M}$ predicts all of the next 50 tokens correctly, i.e. with the highest probability. If it does then we consider tokens $[i : i + 50]$ to be memorized, otherwise not.

Figure C.2 shows what fraction of the 1023 positions with succeeding tokens in the 1024 token string the measure recollects as memorised at each epoch. We consider all positions, because applying this measure in practice involves sampling and investigating all positions shows what fraction of samples of the string would detect memorisation. Figure 13a shows that recall remains essentially zero up until epoch 25, and even at epoch 30 only about 40% of the positions are marked as memorized. Recall is low, because even when the string is largely memorised, models still make a small number of mispredictions, as shown in Figure 13c. Any misprediction, however, results in many substrings not being detected as memorised.

To understand why the string-based memorisation metric severely underestimates memorisation, compare the results in Figure 13a with the accuracy plots in Figure 13b. At epoch 25, roughly 60% of the tokens in the string are predicted correctly, and at epoch 30, around 85% are predicted correctly. By imposing the strict requirement of 50 contiguous correctly predicted tokens, the string-based memorisation metric fails to detect this. Therefore, we argue that memorisation metrics should operate at the token- rather than at the (sub-)string level.

