# OpenReview forum: "Understanding the Mechanics and Dynamics of Memorisation in Large Language Models: A Case Study with Random Strings"
_ICLR.cc/2024/Conference — Submitted to ICLR 2024_

### Official Review · Reviewer_j45i · 2023-10-31

**Soundness:** 3 good
**Presentation:** 3 good
**Contribution:** 3 good
**Rating:** 6
**Confidence:** 2

**Summary:**

This paper explores the mechanics and dynamics of memorization in large language models (LLMs) by training them to memorize random token strings of different lengths and entropies. The study reveals interesting memorization dynamics and investigates the mechanics of memorization, showing that both the local prefix and the global context play a role in token recollection and there is an interplay between them.

I enjoyed this paper. I have several questions in this paper. Overall, I think it is a good experimental design to check the memorization properties.

**Strengths:**

Some strengths of this study include the systematic approach to investigating the mechanics and dynamics of memorization in LLMs, the use of random strings of different lengths and entropies to test the models, and the insights gained into the role of local prefixes and global context in token recollection.

**Weaknesses:**

1) I'm curious if the findings of this paper are more indicative of the underlying structure of Language Models or their training dynamics. The experimental design is intriguing, but I couldn't help but notice that the dataset used doesn't seem to have the scale or correlated content typically associated with "internet-sized" datasets that are used to train Large Language Models (LLMs). Could the absence of such correlations and the relatively smaller dataset size potentially influence the study's outcomes?

2) The paper provides valuable insights, but I wonder if the use of GPT-2, which is smaller than many current LLMs, fully captures the dynamics of "large" language models. While GPT-2 is certainly not a small model, especially when compared to older language models, it seems that the issue of memorization was not as prominent in those older models. Do you think incorporating both smaller and larger models in the study might strengthen the argument that these dynamics are specifically related to the scale of the model? In essence, if the model size does not significantly impact these dynamics within this range, should we still be referring to them in the context of "large" models?

**Questions:**

the same with weakness part

---

> ### Author Response · Authors · 2023-11-17
>
> We thank the reviewer for their feedback and interesting questions! We address them below.
>
> ### Training process and dataset might not be representative of practical settings
> It is true that we only train models to memorize a single random string at a time, a training setup that differs from the large quantities and diversity of training data seen by models during pre-training. Our approach of repeatedly training on the same data is, however, also gaining traction in [recent work](https://arxiv.org/abs/2305.16264), to more effectively utilize data.
> Our choice of training setup is deliberate, as it allows us to ignore confounding factors such as the relative occurrence frequency of some random strings relative to other strings in the training data, and recency and forgetting due to other training data sequences. Studying these factors is potentially also very interesting, but merits its own separate analysis. Recent work by Tirumala et al. (2022) and Carlini et al. (2022) has explored some of these directions.
>
> ### Are the investigated models representative of large language models?
> In the paper, we already study a large variety of model sizes, from 140M to 12B parameters. To test generalization across models even more extensively, we added Llama 2 (7B and 13B) models to the revised paper (see discussion in Section 3.1 and Appendix A.3). Both the papers that introduced [Pythia](https://arxiv.org/abs/2304.01373) (e.g. Figure 4), as well as [Llama 2](https://arxiv.org/abs/2307.09288), report emergent abilities. Therefore, we belive that these models are representative of other large language models.
> In our paper, we observe similar trends across all models, however, larger models are able to memorize faster, and this holds especially for the Llama 2 models. In contrast to the other models, we observe that they skip the guessing-phase completely and go straight to the memorization-phase. They can do this, because they are able to learn the token distribution in the string via in-context learning without additional training (see Appedix A.3). It is also worth noting that the Llama 2 7B model memorizes strings faster than the larger Pythia 12B model, indicating that other factors beyond parameter count also play a role.

---

### Official Review · Reviewer_jCFX · 2023-10-31

**Soundness:** 3 good
**Presentation:** 2 fair
**Contribution:** 2 fair
**Rating:** 5
**Confidence:** 3

**Summary:**

The paper studies how memorization occurs in pre-trained language models. In particular, pre-trained LLMs are further trained to minimize cross-entropy on a dataset of randomly sampled strings, where each character in the string is independently sampled from some fixed alphabet. The main point is that, as these strings are random, vanishing cross-entropy loss can only be explained by memorization during this training, rather than generalizing grammar rules or having seen the string during pre-training. The main result of the paper is that memorization happens in two phases. First, the model learns the marginal distribution of characters in the random strings (i.e. learns the alphabet), and second the model memorizes the strings. The paper contains further results showing that local prefixes of smaller length than the full prefix are sufficient for reconstructing the character at a given position.

**Strengths:**

The paper clearly demonstrates that language models first memorize the token-wise (or in the case of this paper character-wise) marginal distribution before memorizing longer sequences. This gives a clear explanation of the mechanics and dynamics of memorization in transformers, at least for random strings.

**Weaknesses:**

It is quite unclear how practically relevant and/or surprising the observed phenomena are. For example, given $n$ random strings over an alphabet of size $l$, any fixed substring of length larger than $2\log_l n$ is sufficient to uniquely identify each string with good probability by the union bound over all pairs of strings. This is probably sufficient to explain the fact that short prefixes are enough for recovering memorized characters.

On the practical relevance front, the authors write "we argue that our findings call into question an implicit assumption made by existing studies: that memorisation can be reliably detected by checking if the full prefix up to some position in the string can correctly recollect the full suffix following the position (Biderman et al., 2023a)." I do not see how the results in the paper support such a claim. As far as I can tell, this claim is based on the fact that a model may do well at predicting the next token in a random string from a small alphabet by guessing based on having memorized the marginal distribution, even if it has not yet memorized the whole string. This seems unlikely to be the main source of memorization in practical circumstances, where the relevant alphabet size (i.e. number of possible tokens) is quite large. Furthermore, it doesn't even seem to correspond well with the claims in the paper itself, where it is shown that random strings from small alphabets take longer to fully memorize.

**Questions:**

It is possible I am misinterpreting/misunderstanding the main claims relating to the practical relevance of the results in the paper. Is there some more concrete explanation of how these results could inform measurements of memorization? Or more details on how existing measures are making unsupported or misleading assumptions about how memorization should be measured?

---

> ### Author Response · Authors · 2023-11-17
>
> We thank the reviewer for their thoughtful feedback and good questions! We address them below.
>
> ### Some of the results are not surprising
> Being able to uniquely predict the next token from short prefixes is only a *necessary condition* for the recollection accuracy we observe with short prefixes. It’s not clear a priori whether models will converge towards some information theoretically optimal solution. Models might memorize in several ways, and short prefixes are only one of them. Potential alternatives: models might form associations with longer prefixes (e.g. much more than 5 tokens in a 512 token string) as [observed in other settings](https://arxiv.org/abs/2206.04301), or also with tokens not immediately adjacent to the target token (e.g. at the beginning or in the middle of the string), they might learn to detect the number of preceding tokens to do position-based memorization, etc. It's not clear which option(s) models will use without investigation.
> Other findings, such as the initial guessing-phase are also not necessarily expected. Why would a model first learn to associate prefixes with other tokens from the string's alphabet instead of trying to directly memorize the correct token, i.e. directly move into the memorization phase, since the latter reduces the loss even further? We added Llama 2 models to the revision of the paper and observe in Figure 1 (b) that they skip the guessing-phase completely and go straight to the memorization-phase. We show in Appendix A.3 that they are able to learn the token distribution in the string via in-context learning without additional training. This shows that the guessing phase does not happen automatically, but rather depends on the models’ ability to detect distributions via in-context learning.
>
> ### There's insufficient evidence to challenge current notions of memorization
> 1. **Token-level memorization**: We discuss in the global response (3) why detecting memorization at the token-level may be more appropriate than at the string-level.
> 2. **Role of context in associations**: We show that besides the exact prefix of a token (or a string), the context in which it appears also matters for recollection. Therefore, when a model $m$ does not associate string $s$ with prefix $p$, it is not sufficient to conclude that $m$ has not memorized $p | s$, because given the right context $c$, which may not have ever appeared in the training data, $m$ might recollect s given $c | s$. As an example: given the prefix `API_SECRET=`, $m$ might not produce any private information in the training data, and thus one might conclude that the model has not memorized the secret. However, when adding additional context, such as the contents of a file with Python code making an API call to a cloud provider, even if that code has never appeared in that exact form in the training data, it is possible that $m$ might produce an API key for that cloud provider it has seen during its training. This context may be the document in which $p | s$ appeared during training, but it may also be different. Current notions of memorization ignore the context beyond the exact prefix.
> 3. **Memorization/association strength**: Even if a string has been memorized according to existing definitions of memorization, the implications of memorization can be very different depending on how frequently -- i.e. under how many contexts -- a model reproduces it. If a model only produces a memorized string in the context of the exact memorized source document, then the issue (or utility in the case of factual information) is less severe, but if the model produces a secret whenever `API_SECRET=` occurs, under a wide range of contexts, then this is a severe instance of memorization. Notions of memorization should take the strength/severity of memorization into account.

---

> > ### Comment · Reviewer_jCFX · 2023-11-22
> >
> > Thank you for your response, especially the clarifications (and the added section in the appendix) explaining the relationship between your results and existing measures of memorization.
> >
> > I agree that just because it is information-theoretically possible to memorize random strings based on small contexts this does not mean that transformers are guaranteed to do so. However, I do feel that the random string setting is quite idealized, and somewhat far from what we care about in practice. Thus, if there were some very surprising result on LLM memorization in the random string setting, this would be an interesting motivation to study what this implies in the practical setting. However, after reading the other reviews and the corresponding rebuttals, the results for random string setting seem largely unsurprising/predictable, and I don't see any interesting questions to ask about real-world memorization beyond slightly tweaking the metrics used. Notably the Carlini et al. (2022) paper to which you refer proposes another variant of the memorization metric, that is designed to deal with measurement issues that arise in natural language datasets. Based on these issues I am inclined to keep my score.

---

### Official Review · Reviewer_ueBA · 2023-11-02

**Soundness:** 2 fair
**Presentation:** 3 good
**Contribution:** 2 fair
**Rating:** 3
**Confidence:** 4

**Summary:**

The authors propose to study the memorization of pre-trained LLMs on randomly generated strings. They propose to study the memorisation dynamic through training and the influence of the context required to elicit correct predictions. The conclusion of the study, for their very specific setting, is there's a guessing and a memorisation phase, that memorisation happens at token level in no particular order. Furthermore global context is not needed to be kept unchanged, but somehow needs to be preserved, and local context is sufficient.

**Strengths:**

- The paper proposes a toy setting that studies memorisation on LLMs.
- The authors consider a good variety of toy experiments that inspect the training dynamics and the importance of the context
- The presentation is good

**Weaknesses:**

- The results of the analysis may be of littler or no practical usefulness. While the definition of a toy distribution of random strings removes the issue of having to disentangle generalization and memorization, it is unclear to what extent the findings actually reflect real memorization phenomena. This is reflected in the discussions/and implications sections that actually poses the truly interesting questions the authors should have tried to address.
- It would be good if the authors could provide at least a few experiments that correlate their findings on synthetic data with real data.
- The fact models first go through a guessing phase and then start memorising the actual data is unsurprising
- The fact local context is sufficient more than global context may solely depend on the fact the model is autoregressive.

**Questions:**

- The usage of the term dynamics is understandable, but why mechanics? The term may not be the best choice for the subject it refers to.
- Could the authors show experiments on context importance with models that are not autoregressive?
Regarding the discussion section, I would advise the authors to at least carry out some experiments they mention and would validate the usefulness of their analysis in practical settings:
- It would be interesting to at least see some experiments performed on non-random strings or a mixture of random and non-random strings. In that case the random strings would act like canaries that can be more easily detected and cannot be generalised to.
- It would be also more interesting to study whether the influence of global/and local prefixes in recollecting tokens actually does result in stronger reconstruction attacks as suggested.
- On the conclusion that future measures of memorisation should focus on tokens, this may be an issue of the methodology proposed by the authors. That's also why it would be good to provide evidence that quantifying memorisation at higher granularity is not sufficient.

---

> ### Author Response · Authors · 2023-11-17
>
> We thank the reviewer for their time and feedback. We address their concerns below.
>
> ### The random string setting is not representative of natural language
> See global response (1).
>
> ### The existence of a guessing-phase is not surprising
> The initial guessing phase is not necessarily expected. Why would a model first learn to associate prefixes with other tokens from the string's alphabet instead of trying to directly memorize the correct token, i.e. directly move into the memorization phase, since the latter reduces the loss even further? We added Llama 2 models to the revision of the paper and observe in Figure 1 (b) that they skip the guessing-phase completely and go straight to the memorization-phase. We show in Appendix A.3 that they are able to learn the token distribution in the string via in-context learning without additional training. This shows that the guessing phase does not happen automatically, but rather depends on the models’ ability to detect distributions via in-context learning.
>
> ### Memorization in autoregressive/decoder-only models may behave differently from other types of models
> In our paper we deliberately focus on autoregressive models. Autoregressive models are currently the most potent group of language models and are the most important models to study in the context of memorization, since due to their use for text generation they have a high propensity for leaking memorized private information, or for producing inaccurate outputs due to a lack of memorized facts. Many other recent papers also exclusively focus on this group of models. Studying bi-directional contexts would considerably increase the complexity of the analysis and warrant its own separate investigation.
>
> Could you provide us with more details on why you believe that “The fact local context is sufficient more than global context may solely depend on the fact the model is autoregressive”? To us, it’s not clear that the autoregressive nature of models implies a local-context based memorization. In fact, autoregressive models might memorize in several plausible ways, and short prefixes are only one of them. Potential alternatives: models might form associations with longer prefixes (e.g. much more than 5 tokens in a 512 token string), or also with tokens not immediately adjacent to the target token (e.g. at the beginning or in the middle of the string), they might learn to detect the number of preceding tokens to do position-based memorization, etc. It's not clear which option(s) models will use without further investigation, which we provide in this work.
>
> ### The memorization mechanics terminology may not be appropriate
> See global response (2).
>
> ### There's insufficient evidence to challenge current notions of memorization
> See global response (3).

---

### Official Review · Reviewer_LNRg · 2023-11-03

**Soundness:** 3 good
**Presentation:** 3 good
**Contribution:** 3 good
**Rating:** 6
**Confidence:** 3

**Summary:**

This paper focuses on better understanding the mechanism of memorization in LLMs. Basically, the authors tested on tasks of memorizing different random token strings, and observed the dynamics and mechanics of memorization.

There are several interesting observations:
* Memorization has two phases: (1) Guessing-Phase, and (2) Memorization-Phase. The first phase figures out which subset of the alphabet the target string contains. The second phase learns the conditional next-token probability to memorize the target string (within the subset chosen from the first phase)
* During the memorization-phase, memorisation happens at the granularity of individual tokens and not entire strings
* The local context (small number of tokens in the target string) is sufficient to recollect a token at a given position

**Strengths:**

* It is focusing on a timely topic
* The empirical observation has interesting messages

**Weaknesses:**

* It would be great if the authors explain how this observation can give some insight on the training strategy of LLMs. Currently, it looks like the paper has less practical impact.

**Questions:**

.

---

> ### Author Response · Authors · 2023-11-17
>
> We thank the reviewer for their feedback and are glad they appreciate our work!
>
>
> ### Practical impact
> While the random string setting is not representative of many types of data models encountered in practice, we believe that our findings have practical implications for understanding and quantifying memorization in practice.
> For instance, as we discuss in the global response (1), a lot of sensitive data looks -- at least partially -- similar to random strings, such as API and SSH keys, passwords and usernames, phone numbers, URLs, social security numbers, etc. Therefore, memorization of such strings might happen in a similar manner to what we observe in the paper.
> Additionally, we discuss the implications of our results for detecting memorization in practice in global response (3). We argue that practitioners should not a priori assume a specific order of memorization and show that currently used memorization metrics can severely underestimate the degree of memorization of a piece of training data.

---

### Official Review · Reviewer_NLd8 · 2023-11-06

**Soundness:** 3 good
**Presentation:** 3 good
**Contribution:** 2 fair
**Rating:** 5
**Confidence:** 3

**Summary:**

This paper explores the dynamics and mechanics of memorization in causal LMs, where memorization is tested by finetuning pre-trained LMs on random strings. The main results are that (1) memorization occurs in two steps, first fitting to the bag-of-tokens distribution followed by fitting token position; (2) memorizing a token in a sequence doesn't depend on token position; (3) small local contexts are sufficient to greedily generate the memorized token; however, (4) long-range context needs to match the bag-of-words token distribution. Experiments span model sizes (140M -> 12B) and families (Pythia, GPT), as well as length, vocabulary size, and vocabulary entropy of the random string.

**Strengths:**

### Soundness
The methods proposed are technically sound and the results are interesting. The ablations on the influence of global context were well-crafted, where it is found that the token distribution in the global context is important for memorization.

### Presentation
Overall, the paper is written clearly with an easy-to-follow structure/organization, with some minor exceptions (see weaknesses).

### Significance
The methodological framework proposed, including analysis of local/global context, identifying phase transitions in learning, and analyzing order of token memorization, is interesting, easy-to-understand, and can help spark new research in this growing area. Even though the study is done on a very restricted setting, the results demonstrate the usefulness of the _methods_ at teasing apart different stages of memorization.

**Weaknesses:**

## Soundness
Several modeling choices could be better-motivated, and I was unclear on some conclusions. See detailed comments 2, especially 12, 13.

## Presentation
__Major comments__
1. It was unclear to me the relationship between a character and a token (though ultimately it doesn't impact the overall message of the paper) (detailed comments 4, 8, 9).
2. Figures and figure captions (detailed comments 11, 15)

__Minor comments/suggestions (didn't impact the score)__
See detailed comments 1, 2, 5, 6, 10

## Contribution
1. This paper explores memorization in a very restricted toy setting, which the authors allude to in the introduction. Despite motivating the work with applications to, e.g., privacy and factuality for __natural language__, all analysis is done on random strings, an extreme edge case. It is unclear whether the findings will generalize to structured natural language strings, as some results seem to rely on the randomness of the strings (detailed comments 3, 14, 18). Moreover, structured strings are not difficult to generate in a controlled manner using a probabilistic grammar (detailed comment 16). As the focus is on random strings over something more similar to natural language, it is unclear how useful the results are beyond the problem statement defined in the paper. This is my main reservation against acceptance.
2. Results could be better situated in past findings. For instance, Tirumala et al. (2020) also test the effect of model size and dataset size on memorization with similar results. See detailed comments 14, 18.

__Missing reference:__ [Understanding Transformer Memorization Recall Through Idioms](https://aclanthology.org/2023.eacl-main.19) (Haviv et al., EACL 2023)



## Detailed Comments
1. __p1 last paragraph, end of p2:__ In Transformer interpretability literature, _mechanics/mechanistic_ often refers to "mechanistic interpretability", or analysis of how architecture internals like computational circuits lend to behavior. This work does not do that, so the naming may be confusing. Instead, the authors might want to consider diachrony/synchrony or dynamics/statics.
2. __p2 paragraph 1:__ "of different entropies (by sampling tokens at each position..." Larger vocabulary size naturally increases entropy-- to truly disentangle the effect of _entropy_, as suggested, it would be better to keep the same vocabulary size and modulate the token sampling distribution. Alternatively, remove the focus on entropy from this paragraph, replacing with, e.g. "of different vocabulary sizes, which also modulates token entropy, by sampling...".
3. __p2 paragraph 1:__ "Our choice of random strings to study memorisation is deliberate... cannot be explained by other factors such as learning rules of grammar or reasoning." Language Transformers in-the-wild are trained on natural language, which is governed by grammar. In the introduction, the paper's potential impact is framed in the context of naturalistic text (e.g., privacy, factuality, etc); however, this study is restricted to single random strings.
4. __Section 2.1:__ "To create a random string, we first choose an alphabet $A$ that consists of $|A| = l$ unique characters; we call $l$ the length of the alphabet. The alphabet we use for string generation is a subset of a much larger vocabulary of all tokens $V , A \subset V$ ." It is  confusing what a character is. This line implies the alphabet is a subset of tokens in the tokenizer vocabulary.
5. __Section 2.1 last parag:__ "This definition assumes that $\mathcal M$ predicts for position i the token with the largest $P_{\mathcal M}(s_i = t | s[1,i−1]))$." is redundant.
6. __Section 2.1 last parag:__ instead of defining a new term _plurality prediction_ (which is never used again in the article), I would suggest using _greedy prediction_ or _top prediction_ as commonly used in the literature.
7. __Section 2.2:__ "we enforce character-level tokenization... we keep the same subword vocabulary size as the original tokenizer, recognizing that it can impact memorization" This implies that all characters "a-z" are individual tokens in the vocabulary? Enforcing character-level tokenization is confusing given detailed comment 4: does the alphabet consist of individual characters like a-z or individual tokens in the tokenizer vocabulary?
8. __Paragraph Alphabets and string lengths:__ "we focus on alphabets $A$ with $l \in \{2, 4, 7, 13, 26\}$ using the first $l$ letters of the Latin lowercase alphabet, i.e. $A \subset {a, . . . , z}$. We generate random strings of lengths $n \in \{16, 32, 64, 128, . . . , 1024\}$." This can be moved earlier to Section 2.1 to improve clarity.
9. It would be nice to include __examples of the random strings.__
10. __Section 3.3:__ Only in the first parag of section 3.3 is it clear that the LM is only learning one string at a time. Ideally this should be made clear earlier in the paper.
11. __Figs. 3-5___ have GPT2-130M, __Figs 1,2,7__ have GPT2-140M.
12. At __position 0 in Fig. 5a__, the probability is also uniform. Is this given a `<BOS>` token? That is, the probability mass should be spread over the entire vocabulary space and should better reflect the first-token statistics of the pretraining data.
13. __Section 3.3:__ "we conclude that memorisation happens at the granularity of individual tokens and not entire strings." Is this surprising, given that training is via teacher-forcing on a token-level objective?
14. __Section 4.2:__ "at epoch 30 and later, the accuracy of short prefixes, which correspond to less than 5% and 10% of the total string length, is close 100%, which is also the performance of the full prefix when the model has converged. Thus, small local prefixes – much shorter than the entire string – are very effective at correctly recalling tokens." Is this surprising? If the string looks random, then a local prefix is sufficient, as the n-gram will have very low probability. Then, a local prefix will have high mutual information with the next token. This result may not generalize to structured strings containing highly predictable n-grams. This recalls Tirumala et al. (2022) finding that "highly informative" tokens such as numbers get memorized first.
15. __Fig. 8:__ Flesh out the caption to be standalone.
16. __Discussion:__ The discussion on generalising to non-random strings could be expanded. Virtually all applications of language Transformers apply to structured strings (natural language). E.g., rather than define the distribution over the _alphabet_, the distribution over strings can be defined using a _probabilistic grammar_.
17. __Discussion:__ "For instance, to infer whether a token string is a member of the training data, it may be sufficient to infer that some unique part of the string has been memorised." The results show that memorization happens on a token-by-token basis-- then, does token-level memorization really tell you anything about the string-level? E.g., if you prove that the word "The | `<BOS>`" has been memorized by the model, then does that prove that "The sky is blue" has been memorized?
18. The result that token positions are equally likely to be memorized first may also be a consequence of using random strings, as the n-gram probabilities are the same. As soon as you move to non-random strings, this result may break down due to the statistics of natural language. For instance, Tirumala et al. (2022) find differentiated memorization speed according to POS.

**Questions:**

See detailed comments 4, 8-9, 12-14, 17

---

> ### Author Response · Authors · 2023-11-17
>
> We thank the reviewer for their very detailed and thoughtful feedback! You are raising great points which are very helpful to improve the paper! We address them below.
>
> ### Clarifying the string construction process
> The vocabularies of the tokenizers used in this paper contain both tokens for individual characters, e.g. “a”, “b”..., as well as tokens spanning multiple characters, e.g. “abs”. By “character-level tokenization” we mean that we only use tokens corresponding to individual characters, not character sequences. To make this part clearer, we updated the terminology in the string construction process in the revised version of the paper. We refer to the elements of the alphabet as tokens and only use the term character if the alphabet tokens indeed correspond to the characters in the Latin alphabet.
>
> ### The random string setting is not representative of natural language (DC 3, 14, 18)
> See global response (1).
>
> ### We should use structured synthetic data, i.e. probabilistic formal grammars (DC 16)
> Using formal (probabilistic) grammars to create structured data is a great way to investigate how models learn to reason according to the grammar rules, and we are exploring this direction. We believe, however, that it is not directly suitable for understanding memorization. Intuitively, for a string to be memorized, a model must have seen and remembered it from its training data, there should not be any other way to generate it. We can achieve this with random strings, but if we generate data in a structured manner, models might produce valid strings by learning and using grammar rules, instead of memorizing the training strings.
>
> ### Missing discussion of related work
> Thank you for pointing out the relevant related work and the connection between our findings and those of Tirumala et al. (2022). We added the missing reference in the revised paper.
>
> The work of Tirumala et al. (2022) and their findings about memorization order are indeed relevant in light of our observations about random memorization order. Their high-level message that different parts of text can be memorized at different speeds, and that memorization order is thus hard to predict matches our findings. However, we are skeptical about the soundness of some of their claims on memorization order. Their definition of memorization is problematic, since it considers any correctly predicted token $y$ following a prefix $s$ as memorized. However, it is likely that at least some correct predictions do not happen because of memorization, but rather because of other reasoning abilities that models are acquiring. Take for example "This morning I drank a", where models might correctly predict "coffee" without every having seen this sentence in their training data. Since their findings on differences in memorization order for different POS are based on this definition, we would approach them with caution and do not include a more extensive discussion.
> The difficulty of disentangling memorization and predictions made using other mechanisms is the core reason for restricting ourselves to random data in this paper, because in random strings, correct predictions can only be attributed to memorization, since there are no rules to derive the next token from. We would like to point out, though, that we think studying memorization at the token-level, rather than string-level, is more appropriate (see global response (3)), but inferring memorization from just a single correctly predicted token seems highly optimistic. A more robust measure should check whether a reasonably large fraction of tokens in a sufficiently long string postfix are correctly predicted.
>
> ### Results on short prefix recollection are not surprising (DC 14)
> Being able to uniquely predict the next token from short prefixes is only a *necessary condition* for the recollection accuracy we observe with short prefixes. It’s not clear a priori whether models will converge towards some information-theoretically optimal solution. Models might memorize in several ways, and short prefixes are only one of them. Potential alternatives: models might form associations with longer prefixes (e.g. much more than 5 tokens in a 512 token string) as [observed in other settings](https://arxiv.org/abs/2206.04301), or also with tokens not immediately adjacent to the target token (e.g. at the beginning or in the middle of the string), they might learn to detect the number of preceding tokens to do position-based memorization, etc. It's not clear which option(s) models will use without further investigation, which we provide in this paper.

---

> ### Author Response · Authors · 2023-11-17
>
> ### Unclear how to relate token- to string-level memorization (DC 17)
> It is true that in a natural language settings, simply predicting a single token correctly, given some prefix from the training data, does not imply that that token (or a larger string that it is contained in) has been memorized, since the model could also predict that token using word statistics, grammar rules, etc. (e.g. “blue” in “The sky is blue.”). It is merely a necessary condition for memorization, but not a sufficient one. Current approaches to quantifying memorization using string-based notions (Carlini et al. (2022)) assume that if a sufficiently long string from the training data is predicted exactly, given its prefix, then it is very unlikely that this is due to something other than memorization (because predicting each token exactly as in the training data independently is highly improbable). By focussing on random strings in this study, we sidestep this issue. If models can predict the next token correctly, it has to be because of memorization.
> We discuss the implications of our findings for detecting memorization in the global response (3).
>
> ### The memorization mechanics terminology may not be appropriate (DC 1)
> See global response (2).
>
> ### Other detailed comments
> 2. Disentangling entropy: To make sure that the behaviour we observe is indeed a result of the strings’ entropy, we conduct a control experiment in Appendix A.2 (mentioned in Section 3.1). We modify the entropy of strings with fixed alphabet size (26) by increasing the probability of sampling the first character (“a”) such that the generated strings have the same entropy as the strings with smaller alphabet size. Our results show the same patterns for different entropy levels at the same alphabet size as for different entropies due to different alphabet sizes. Due to space constraints, we could not include these results in the main paper.
> 5. We removed this part in the revised version.
> 6. We switched to using greedy decoding terminology.
> 9. We added examples with different alphabet sizes and entropies in Appendix A.4.
> 10. In most cases we train models only on single strings, but in Figure 8 we also train them on multiple strings at a time, so this depends on the specific experiment.
> 11. Good catch, 140M is correct, we corrected Figures 3-5 in the revision.
> 12. We do not use <BOS> tokens for GPT-2 and Pythia models. Figure 5a simply shows whether the token at the respective position has been predicted correctly at that timestep. The first column shows the prediction for the second token in the string, since the model does not produce a prediction for the first token (the output for the first token is the prediction for the second token). Please let us know if this doesn’t clarify the question.
> 13. We are driving the loss to zero over all tokens in the string at the same time. Alternative memorization dynamics are therefore also conceivable, such as memorizing tokens in chunks, from left to right, etc. We discuss the implications of this finding in the global response (3).
> 15. We moved details and takeaways into the Figure caption.

---

> ### Comment · Reviewer_NLd8 · 2023-11-21
> **Thanks for the response**
>
> Thanks for the response! The paper is a lot clearer now. I will retain my score (explanation below), but won't oppose acceptance if other reviewers champion for it.
>
> All of my remarks regarding clarity/presentation have been resolved. The remaining doubt I have is still the one on whether results on random strings will be relevant to the community. Re: the point on probabilistic formal grammars ("We can achieve this with random strings, but if we generate data in a structured manner, models might produce valid strings by learning and using grammar rules, instead of memorizing the training strings."), formal grammars allow for primitives that are more than one character long, which still lets you have smaller random substrings within the string.
>
> An example:
>
> e.g.
>
> S | A B
>
> A | [random string sampling one subset of the vocab]
>
> B | [random string sampling another subset of the vocab]`

---

### Author Response · Authors · 2023-11-17

We thank all reviewers for their time and feedback, which is valuable to help us improve the paper! We address common issues below and specific comments in the responses to each reviewer.

### Changes in the revised version of the paper
We uploaded a revised version of the paper to address some of the issues that were raised. It contains the following changes:
- We clarified the token vs character terminology and revised the description of the data-generation process accordingly in Section 2.
- We included results on Llama 2 models in Section 3, as well as a discussion of their behaviour using observations about in-context learning in Appendix A.3.
- We added a brief rationale for using the term mechanics in Section 4.
- We added results in Appendix C.2 showing that string-based memorization metrics like the ones in Carlini et al. (2022) can severely underestimate the amount of memorization in a string.
- We added a missing citation in Section 1.1.
- We added examples of the random strings used in the apper in Appendix A.4.
- We moved more of the description of Figure 8 into its caption.

### (1) The random string setting is not representative of natural language
We conduct our experiments on random strings, because they i) guarantee that models have not seen the strings during pre-training, ii) ensure that models have to memorize the data in order to achieve low loss, and iii) give us precise control over all aspects of the data, such as string length, alphabet size, and entropy. Achieving all of these properties with natural language would not be possible. We believe that understanding memorization of random strings is valuable in its own right, for the following reasons:
1. **Sensitive data is often similar to random strings**: A lot of sensitive data looks -- at least partially -- similar to random strings, such as API and SSH keys, passwords and usernames, phone numbers, URLs, social security numbers, etc. Therefore, memorization of such strings might happen in a similar manner to what we observe in the paper.
2. **Preliminary results on natural language data suggest similar trends**: Prompted by our findings on random strings, we also conducted follow-on experiments on natural language data, by training models to memorize news articles not published yet when the models were trained. In many cases, we can make similar observations as for random strings. For instance, larger strings take longer to memorize, larger models memorize faster, there is no directly discernible order of memorization in the string (though we did not test for Tirumala et al (2022)’s findings yet), and short prefixes are sufficient to recall many of the tokens correctly, though fewer than in random strings (5 token prefixes recall ~95% vs ~90% in ~128 token strings, ~80% vs 65% in ~512 token strings, in random vs natural language strings). Some other observations differ, as expected, e.g. we do not observe a distinct Guessing-Phase (probably due to the much larger and more sparsely used token vocabulary), and some are not straightforward to test, e.g. the role of alphabet size and entropy on memorization speed. Overall we believe that memorization in natural language data follows broadly similar trends, but more research is needed to get conclusive evidence.
3. **Memorization of natural language data warrants its own paper(s)**: Cleanly understanding memorization in natural language data requires solving a number of challenging problems, such as distinguishing between tokens correctly predicted due to memorization vs reasoning using grammar and other rules and determining whether substrings of any strings used for memorization might have appeared in the training data. One would also want to study the impact of other dimensions on memorization than the ones relevant for random strings, such as e.g. the type of PII data (phone numbers, email addresses, etc.) vs alphabet size and entropy. We already have more interesting observations on random strings than can comfortably fit into a single paper. To include results and methodology for natural language, we would have to omit important results on random strings. Therefore, memorization of natural language strings very much warrants its own, separate paper(s) and we will keep this paper focussed on random strings. We believe, however, that our insights from random strings will be valuable for finding interesting research questions in this space. Our methodological approach can serve as a template for research on memorization in natural language data, as well for learning on structured data by using formal grammars, for instance.

---

> ### Author Response · Authors · 2023-11-17
>
> ### (2) The memorization mechanics terminology may not be appropriate
> We call some of our findings mechanics, because (similar to physics) they can be used to explain and predict the dynamics, e.g. differences in memorization speed for strings of the same length and different alphabet sizes because models need to learn associations with prefixes of different lengths, different splits of the same string having almost the same memorization speed, because models learn the same prefix mappings, etc. We argue that you can model the mechanics of a system at multiple levels (e.g., classical vs. statistical vs. quantum mechanics in physics). In this work, we are concerned with the mechanics at the level of (latent) mappings learned by the LLM, i.e. based on how the outputs of models correspond to their inputs, in the context of random string memorization. We show distinct roles that the exact prefixes and the remaining context play. In addition to the input-output level dynamics, there are also lower-level dynamics, which is often what mechanistic interpretability is concerned with. An analogy for multi-level dynamics in physics would be the mechanics of a spring. Its behaviour can be modelled and explained at a high-level (how much does some amount of force stretch the spring) or at an internal level (how is the atomic lattice structure of the sprint deformed by applying the force) and also at the level of quantum-mechanics (how do atoms in the sprint interact). All of these are different levels of studying mechanics of the same object. We focus on the higher-level mechanics here.
>
> ### (3) Why studying memorization at the token-level (rather than the string-level) may be more appropriate
> Our results caution against a priori assuming specific memorization patterns. Prior work on quantifying memorization (Definition 3.1 in Carlini et al (2022)) implicitly assumes memorization in a block contiguous structure. Such a definition may be too strict and may fail to detect largely memorized strings, even if just a few tokens within a string are not fully memorized. We include experiments in the revised version of the paper in Appendix C.2 that show that, indeed, the aforementioned string-based metric only detects a small fraction of substrings of almost completely memorized 1024 random token strings as memorized. Detecting memorization at the token-level provides a better picture of the actual degree of memorization.
> A simple way of using our insights to improve the detection of memorized strings in practice therefore would be to count the number of (not necessarily adjacent) correctly recollected tokens in a string, given prefixes from the same string. The higher the fraction of correctly predicted tokens, the more likely it is that the model has memorized (parts of) the string. This may be more robust than requiring that the entire string is reproduced correctly by the model without gaps.

---

### Meta-Review · Area_Chair_kfTS · 2023-12-04

**Metareview:**

This paper studies memorization in causal LMs by finetuning a pre-trained model to learn random strings. The results show a two-phase learning dynamic, where first the token distribution is learned, and then token position is acquired. They uncover moreover that small prefixes suffice to generate memorized strings.

The issue of how LMs memorize information is of theoretical and practical importance, and the experiment is well-designed and easy to follow in its simplicity.

The main concern raised by the reviewers is whether studying learning of completely random strings in a controlled, small-scale setup in which the model is only exposed to one type of training data is indicative of what happens in large LM training, where memorization and generalization are strongly entangled. This is a difficult conundrum: on the one hand, if we don't simplify the problem, we cannot assess the impact of different factors. On the other hand, if we simplify, it's not clear we are studying the same problem anymore.

Indeed, the fact that, in the experiments with the more state-of-the-art Llama 2 models, the two-phase pattern disappears casts doubts on the generality of the results.

I would consider this a small but potentially useful contribution, but a majority of reviewers, even after rebuttal, do not think the article has sufficient potential impact to appear at ICLR in its current version.

**Justification For Why Not Higher Score:**

A majority of reviewers expressed a strong concern about the general implications of the study.

**Justification For Why Not Lower Score:**

N/A

---

### Decision · Program_Chairs · 2024-01-16

Reject